# OpenReview forum: "Towards Effective Discrimination Testing for Generative AI"
_ICLR.cc/2025/Conference — Submitted to ICLR 2025_

### Official Review · Reviewer_79Lv · 2024-10-31

**Soundness:** 3
**Presentation:** 4
**Contribution:** 3
**Rating:** 8
**Confidence:** 4

**Summary:**

This paper discusses the legal and technical literature related to discrimination in Generative AI. It focuses on four main challenges: (1) the mismatch of traditional fairness metrics, (2) the reliance on red teaming, (3) evaluating fairness after different interactions and (4) impact of changes in hyper-parameters. The authors provide evidence of each challenge by running experiments on different datasets, some textual and some images and provide recommendations for how best to assess discrimination arising from GenAI in relation to non-discrimination laws and AI-specific laws and regulations.

**Strengths:**

I enjoyed reading this paper - it is clear and well written. Further, it provides a good narrative of existing literature and collates four major challenges in assessing discrimination of generative AI. The experiments are sound, well thought out, and give evidence for the issues identified. Thus, the motivation for the provided recommendations on how best to assess discrimination are clearly motivated. This is a highly topical and important field which needs urgent consideration. Mapping legal concepts to technical concepts is the main novelty of this paper.

**Weaknesses:**

Although the recommendations are well supported, they are general, not very actionable and are thus not a great addition to the paper. The abstract claims the recommendations are 'practical' which I think is a bit misleading. For example, in the "Mitigation" part of Section 4.1 (line 295 onwards) the following recommendations are made:
- "fairness researchers should attempt to create metrics and testing regimes that shed light on how GenAI behavior may influence decision-makers' perceptions of candidates from different demographic groups. One approach could involve developing standardized frameworks that measure bias..."
- "By considering this larger, tailored suite of metrics..."
I do not think these recommendations go far enough in being practical, more considerations of future work that should be explored.

The work is very topical thus there has been a lot of testing done for discrimination in GenAI already, the originality comes in the narrative and legal perspective however this legal perspective is heavily focused on the U.S. legal landscape.

Other:
- Introduce all terms or do not use them (e.g., frontier models, NLP)
- Figure 1 doesn't add much for me, I can see it maps directly to the sections/issues but it would be useful to use the same terminology in the diagrams or in the section headers to relate it directly

**Questions:**

- You mention liability (and accountability) a few times throughout the paper - do you have any thoughts on how this intertwines with the legal landscape? As stated, this is a challenge particularly in GenAI, could be useful to discuss this a bit more, or reference discussions on it.
- In line 142, you state that when GenAI is used to make allocative decisions in a way that mirrors traditional decision making... Could you expand on this? What is an example and how does it mirror? Further in the paper this is explored further but it could be good to provide and example here. Has it actually been used in a real-life scenario?
- Related to the above: the paper notes that "the experiment for applying traditional fairness notions to GenAI systems is high-fidelity of a real hiring application to demonstrate bias evaluation and downstream discriminatory behaviour". This is of course necessary for the paper, but do you know any examples of real hiring applications using GenAI?
- Do you have any thoughts on how the application focused approach of the EU AI Act connects to the assessment of discrimination of GenAI?

---

> ### Author Response · Authors · 2024-11-19
> **Author Response**
>
> We thank the reviewer for their consideration of and feedback on our submission.  We are pleased to hear that they found the paper to be informative and well-written.  Please see below for responses to specific questions and comments.
>
> **Although the recommendations are well supported, they are general, not very actionable and are thus not a great addition to the paper. The abstract claims the recommendations are 'practical' which I think is a bit misleading. For example, in the "Mitigation" part of Section 4.1 (line 295 onwards) the following recommendations are made:**
> - **"fairness researchers should attempt to create metrics and testing regimes that shed light on how GenAI behavior may influence decision-makers' perceptions of candidates from different demographic groups. One approach could involve developing standardized frameworks that measure bias..."**
> - **"By considering this larger, tailored suite of metrics..." I do not think these recommendations go far enough in being practical, more considerations of future work that should be explored.**
>
> We appreciate the reviewer’s concern.  We are limited by space, and ultimately we feel that very specific suggestions for each issue would require extensive experimentation beyond the scope of this paper; it is precisely this work that we hope to spur with our paper.  However, we do offer the following practical suggestions corresponding to each case study:
> - Use metric suites tailored to specific applications (hard to specify more, given the context-dependence).
> - Ensemble red teaming results across different methods and parameter choices
> - Perform tests that resemble, as closely as possible, the interaction mode in deployment
> - Evaluate safety across a range of free hyperparameters.
>
> In response to the reviewer’s concern, we have expanded the subsection on Mitigation in Section 4.1 to be more specific about how our suggestions might be implemented in this particular hiring context.
>
> **The work is very topical thus there has been a lot of testing done for discrimination in GenAI already, the originality comes in the narrative and legal perspective however this legal perspective is heavily focused on the U.S. legal landscape.**
>
> We have added additional comments on non-US policy efforts in Section 3 (Line 141), and appendix B.
>
> **Introduce all terms or do not use them (e.g., frontier models, NLP)**
>
> We appreciate this concern, and have removed these terms.
>
> **Figure 1 doesn't add much for me, I can see it maps directly to the sections/issues but it would be useful to use the same terminology in the diagrams or in the section headers to relate it directly.**
>
> To address this, we have updated Figure 1 to more clearly refer to the case studies in terms of issue ordering and the correspondence between descriptions in the figure and section headers.

---

> > ### Author Response · Authors · 2024-11-19
> > **Author Response (continued)**
> >
> > **You mention liability (and accountability) a few times throughout the paper - do you have any thoughts on how this intertwines with the legal landscape? As stated, this is a challenge particularly in GenAI, could be useful to discuss this a bit more, or reference discussions on it.**
> >
> > Liability in AI systems is particularly complex because the development and deployment processes are often separate. Developers create the systems, while users or deployers integrate them into real-world applications, often with limited understanding of the underlying mechanics or data. We attempt to summarize some discussion of these points here in the response, but provide more discussion in Appendix B.
> >
> > Historically, discrimination law has primarily focused on the entities using or deploying systems, holding them accountable for discriminatory outcomes and decisions. In contrast, other legal frameworks, such as product liability, have centered on developers or manufacturers of products. For AI systems, and particularly for GenAI, the emerging approach is to distribute liability across both developers and deployers, sometimes with different requirements. For instance, the EU AI Act includes provisions that apply to both developers and users of GenAI/AI systems– for example, Article 52 outlines requirements for general-purpose AI providers to conduct risk assessments, implement mitigation measures, and ensure transparency, regardless of the specific application for which the AI is eventually used, and Article 29 requires deployers of all high-risk AI (not only GenAI) systems to monitor the operation of their systems. It is worth noting that the proposed EU AI Liability Directive, which is under negotiation, leans more heavily toward addressing developer accountability, particularly where defects in the system’s design or training contribute to harm. However, the Directive does not exclude users from liability when users directly violate discrimination laws.
> >
> > In the U.S., liability for discriminatory outputs of GenAI systems is typically addressed through a patchwork of domain-specific laws, which apply in contexts like employment, lending, or housing. These laws generally hold users or deployers responsible for discriminatory practices, regardless of whether those practices result from an AI system. However, recent litigation highlights the evolving application of anti-discrimination law to AI technologies that shifts more burden to the developer. In a notable case, the U.S. Equal Employment Opportunity Commission (EEOC) supported a lawsuit against Workday, a developer—not a deployer—of an AI system, alleging that its AI-powered job application screening tools disproportionately disqualified candidates based on race, age, and disability. A federal judge allowed the proposed class-action lawsuit to proceed, emphasizing that Workday’s tools could be viewed as performing tasks traditionally associated with employers and were therefore subject to federal anti-discrimination laws. This case illustrates that developers can face liability, and it highlights the often-blurred lines between developers and deployers. Similarly, New York City’s AI bias audit requirement for hiring tools (Local Law 144) places obligations on deployers to audit and disclose information about tools they may not have developed. Our analysis provides yet another reason to not view this distinction as straightforward, given that harm can arise from a user’s specific implementation or customization of the AI system. Importantly, this is an ongoing issue and, as we note in the paper, AI researchers can add to the conversation by creating more ways for developer-deployer collaboration on discrimination testing.

---

> > > ### Author Response · Authors · 2024-11-19
> > > **Author Response (continued)**
> > >
> > > **In line 142, you state that when GenAI is used to make allocative decisions in a way that mirrors traditional decision making... Could you expand on this? What is an example and how does it mirror? Further in the paper this is explored further but it could be good to provide and example here. Has it actually been used in a real-life scenario?**
> > >
> > > We attempt to clarify and welcome follow-up questions: by mirroring traditional decision-making, we mean the situation when a GenAI system is the entity that makes the final call on who gets an opportunity or resource (who is hired? who gets a loan? who gets approved to rent a house?). In other words, the GenAI makes a classification decision (e.g. to hire or not to hire) and that is the final decision that the company goes with. In this case, traditional anti-discrimination frameworks can apply to enforce testing of disparate impact in the GenAI system’s decisions (e.g. see if it hires more men than women, for example). This mirrors traditional decision-making in hiring, where a person would make a final decision on who to hire and not to hire based on information they receive (or potentially, some deterministic process leads to a hiring decision based on a questionnaire or verification of qualifications), in that the output of the GenAI is a final, 0/1 decision of whether to hire someone. We note that this setup is not common in lines 145-148. Instead, what we do have evidence of is GenAI systems providing indirect influence on the hiring process, e.g. by providing a summary of a resume, as we outline in our first case study.  Please let us know if we can clarify any further.
> > >
> > > **Related to the above: the paper notes that "the experiment for applying traditional fairness notions to GenAI systems is high-fidelity of a real hiring application to demonstrate bias evaluation and downstream discriminatory behaviour". This is of course necessary for the paper, but do you know any examples of real hiring applications using GenAI?**
> > >
> > > The set-up in the experiment mimics what we believe is a more common use of AI in hiring: namely, that a GenAI system summarizes resumes and a human makes the final decision (though in this case we model a human by another more advanced GenAI system). While it’s difficult to find examples of particular companies openly admitting to use GenAI systems to screen candidates resumes, articles we cite in the paper says that “dozens of HR trade blogs have talked up the potential of using it to automate certain HR tasks, including analyzing resumes and assessing applicants’ skills” (Bloomberg 2024) . Further, there are several companies that have formed recently offering variations on resume summary as a service, suggesting that it is a growing trend: for example, LinkedIn’s Recruiter2024 apparently works by a recruiter inputting a prompt such as “I want to hire a senior growth marketing leader” into a LLM-based tool, and the tool will search through and evaluate linkedin profiles to find high-quality candidates. While this description is vague, we assume this process includes summarize the qualifications listed on their profile (essentially a resume) and determining which profiles may be the best fit for the position. We hope this helps answer your question.

---

> > > > ### Author Response · Authors · 2024-11-19
> > > > **Author Response (continued)**
> > > >
> > > > **Do you have any thoughts on how the application focused approach of the EU AI Act connects to the assessment of discrimination of GenAI?**
> > > >
> > > > We understand the question to ask whether the AI Act’s risk-based approach to regulation, whereby regulatory scrutiny depends on the risk of the domain in which an AI tool is used, also applies to GenAI. This is an interesting and complex question given that the EU AI Act was drafted before the current attention on GenAI. We discuss this question here, and have added this discussion to Appendix B, and also added more comments on non-US legal approaches to Section 3 of the paper.
> > > >
> > > > The EU AI Act indeed adopts a risk-based approach, classifying AI systems into four categories: prohibited, high-risk, limited risk, and minimal risk. Initially, the Act was primarily tailored to traditional AI applications like credit scoring, recruitment, or healthcare. However, as GenAI gained prominence during the drafting process, it was explicitly incorporated through amendments to address its unique challenges. Specifically, the Act was expanded to include general-purpose AI (GPAI) systems, such as GenAI, within its scope. These systems often serve as foundational models that can be fine-tuned or customized for specific applications across diverse domains.
> > > >
> > > > To the extent that a GenAI system is used like a traditional AI system—meaning for a specific use case—the risk-based approach would likely apply. For example, if a GenAI system was used to provide credit scores to borrowers it would likely be classified as high-risk and the Act’s Articles related to high-risk systems would apply. However, unlike traditional AI high-risk systems that are typically tied to specific domains, because GenAI models often produce outputs that often do not map directly onto allocative decisions, the EU AI Act creates rules specific for GenAI. To address this, the Act makes a distinction between GPAI systems that have systemic risks and those that do not, tailoring specific provisions to each category. For GPAI systems that pose systemic risks, Article 52 introduces additional requirements, such as the obligation of developers to conduct comprehensive risk assessments and implement mitigation strategies to address risks. For GPAI systems without systemic risks, the obligations are less stringent but still require developers to ensure that their systems are designed transparently and include mechanisms to minimize foreseeable risks, such as Article 54 which creates a documentation requirement.
> > > >
> > > > In short, the risk-based approach of the Act continues to apply to GenAI when deployed in a specific setting covered. But the Act goes beyond the core requirements for GenAI, creating a systemic/non-systematic risk distinction rather than is risk-based categories used primarily for traditional AI systems.

---

> > > > > ### Comment · Reviewer_79Lv · 2024-11-25
> > > > >
> > > > > Thank you for the thorough response and the changes in the paper. I think these additional discussions are useful. The comments on the EU AI Act are especially interesting to me. I maintain my high score as I think the paper provides a useful investigation into discrimination in GenAI. For the future, the paper could be improved by the recommendations being more specific, novel and actionable.

---

### Official Review · Reviewer_LRTp · 2024-11-05

**Soundness:** 2
**Presentation:** 3
**Contribution:** 2
**Rating:** 5
**Confidence:** 3

**Summary:**

The paper discusses the gap between LLM evaluations and regulatory goals in generative AI systems. The authors present four case studies demonstrating how current fairness evaluation methods may fail to prevent discriminatory behavior in real-world deployments. The paper makes connections between technical and legal literature while offering practical recommendations for improving discrimination testing.

**Strengths:**

The paper discusses a timely an important topic, and some of the case studies are quite compelling (in particular the ones from 4.2 and 4.4). As far as I can tell, the main originality of the paper lies in the connection to the legal literature, and the case studies showcasing certain important phenomena, which make the assessment of the fairness of models difficult or legally challenging.

I think the paper is quite clear, and may be somewhat significant for an audience that is not particularly versed with LLMs.

I thought the second case study was well executed, and showcases an important phenomenon that practitioners are somewhat aware of, but it's good to have a comprehensive demonstration of. I also though the 4th case study does a good job at highlighting one aspect of deployment that people don't often think about (generation hyperparameters) and raising interesting legal questions about who should be responsible for harmful behaviors that are due to changes in hyperparameters like that.

**Weaknesses:**

Despite the fact that the phenomena being showcased are important, I don't think most of them are novel: I expect that most practitioners would not find any of the case studies surprising, and would already know most of the phenomena being discussed.

Let's take for instance the first case study: the authors make the case that using quality metrics like ROGUE will not necessarily correlate with fairness. I don't think this would surprise anyone: this is exactly what spurred many evaluations that are specifically targeted for fairness, toxicity, and so on. In the technical reports for frontier models, there are plenty of such evaluations being made (which again indicates that something quite similar to the main takeaway from case study 2 is already mainstream: i.e., people should be doing many different evals with different setups, and looking at the whole picture).

I would have found the first case study much more compelling if it were using some of the state of the art benchmarks for LLM fairness. Moreover, I found it a bit suspicious that the authors used Llama 2 for the first case study, but not for the later ones. This made me question whether that model (together with the others) was cherry-picked to make the point they were trying to make – that's fine (because ultimately you're just trying to show that this kind of phenomenon _can_ happen), but it does detract from the impact of your point, as it makes one question how common this kind of phenomenon actually is.

Moreover, the first case study suggests to use alternate fairness metrics, showing that they are more successful. This feels a bit to me like re-inventing the wheel, as there are already other evals which are specifically tailored for measuring fairness, and my guess would be that such evaluation scores are already more correlated with actual fairness of outcomes in the setting that is tested. Another smaller qualm with the setup is that there were no error bars reported in Figure 2, and the number of resumes generated was quite small, when considering that they are AI generated, so the additional generation costs are minimal. This makes it hard to assess the significance of results.

Despite the second case study being well executed, the takeaways are a little underwhelming – as mentioned above, I think they are already the standard practice for any entity that knows what they're doing.

I had trouble understanding the third case study, especially the experimental setup. How is the interaction history created exactly? Do you have an example of what an interaction looks like somewhere?

**Questions:**

Already asked above.

---

> ### Author Response · Authors · 2024-11-19
> **Author Response**
>
> We thank the reviewer for the time and care taken in reviewing our submission.  We are encouraged that the reviewer felt that our research covers a timely and important topic, and that our case studies are compelling and able to showcase the gaps between the legal and technical aspects of anti-discrimination efforts.  Below, we respond to particular concerns that the reviewer has raised.
>
> **Despite the fact that the phenomena being showcased are important, I don't think most of them are novel: I expect that most practitioners would not find any of the case studies surprising, and would already know most of the phenomena being discussed.  Let's take for instance the first case study: the authors make the case that using quality metrics like ROGUE will not necessarily correlate with fairness. I don't think this would surprise anyone: this is exactly what spurred many evaluations that are specifically targeted for fairness, toxicity, and so on. In the technical reports for frontier models, there are plenty of such evaluations being made (which again indicates that something quite similar to the main takeaway from case study 2 is already mainstream: i.e., people should be doing many different evals with different setups, and looking at the whole picture).**
>
> We agree with the reviewer that the empirical results presented in Sections 4.1-4.4 may not surprise an ML researcher deeply familiar with the relevant literature.  However, as noted in our main rebuttal, we do believe there is meaningful novelty in our work exploring LLM-assisted hiring applications, variability of red-teaming results, the effects of multi-turn interactions on fairness and toxicity, and the effects of hyperparameter choices on text to image bias.  This novelty supports our main goal of showing how gaps between existing and emerging regulation and current fairness techniques can lead to deployment of reportedly fair, yet actually discriminatory GenAI systems, and we believe this makes these results significant and interesting for the ICLR audience.
>
> In response to the reviewer’s comments on the first case study specifically, we note that we are not making claims about a specific metric. The novelty in the hiring example derives from the study of an application where the implications of bias in an LLM’s output summaries cannot be easily prevented or understood under existing law and evaluation techniques. We are showing that the flexibility that exists in fairness evaluation, plus mismatched areas of focus and lack of specificity in regulation, means that a developer could perform a fairness test that satisfies testing requirements in emerging (Gen)AI regulation that still admits a biased decision-making system (i.e., one that exhibits disparate impact across demographic groups in its decisions).
>
> In particular, we do not evaluate the claim that ROUGE will correlate with fairness, but that equalizing common LLM/GenAI performance metrics— in this case, ROUGE—may not correlate with actually preventing traditionally understood forms of discrimination (i.e. disparate impact) in a GenAI system’s downstream use case. Equal model performance across demographic groups is a common fairness testing paradigm [3], and we argue is a reasonable fairness test even for those who may know what they’re doing. However, as we see, it does not correlate well with preventing disparity in the downstream hiring task because despite equal summarization accuracy, the resumes admit differences in sentiment, length, etc. which is contextually relevant to the downstream task and results in disparate rates of selection across demographic groups. This disconnect is a problem especially in areas covered by disparate impact law such employment, credit, and housing— GenAI systems may be contributing to downstream disparate impact but this may not be caught between the traditional and emerging legal frameworks.
>
> With respect to frontier model technical reports:
> - Llama-3 technical report [1] uses the ROUGE-L metric
> - GPT-4 technical report [2] contains no results on summarization
>
> Thus, we believe that they would not be useful for crafting an evaluation protocol to anticipate how an LLM’s behavior might impact outcomes in many important emerging applications, such as the one shown in this case study.  In general, the Llama-3 technical report makes no mention of the words fairness or discrimination, and the GPT-4 report only mentions that they consider these issues, but offer no open protocols or datasets for others to adopt.  Thus we believe that such reports offer no useful guidance on overcoming any of the issues pointed out in our work.
>
> - [1] The Llama 3 Herd of Models https://arxiv.org/pdf/2407.21783
> - [2] GPT-4 Technical Report https://arxiv.org/pdf/2303.08774
> - [3] Verma, Sahil, and Julia Rubin. "Fairness definitions explained." https://dl.acm.org/doi/pdf/10.1145/3194770.3194776

---

> > ### Author Response · Authors · 2024-11-19
> > **Author Response (continued)**
> >
> > **I would have found the first case study much more compelling if it were using some of the state of the art benchmarks for LLM fairness. Moreover, I found it a bit suspicious that the authors used Llama 2 for the first case study, but not for the later ones. This made me question whether that model (together with the others) was cherry-picked to make the point they were trying to make – that's fine (because ultimately you're just trying to show that this kind of phenomenon can happen), but it does detract from the impact of your point, as it makes one question how common this kind of phenomenon actually is.**
> >
> > We do agree with the reviewer that it would be interesting to explore whether other technical notions of fairness might be correlated or not with these context-dependent outcomes, but we believe such studies are outside the scope of our paper.  However, as we argue and demonstrate throughout the paper, it cannot be taken for granted that the latest research techniques are guaranteed to achieve specific real-world outcomes.
> >
> > Regarding the inclusion of Llama-2 in the first experiment, we would note that it offers the best summary quality according to ROUGE among a set of similarly sized models, and thus seemed reasonable for inclusion in this example.  In recognition of this concern of experiment consistency, we have added Llama-2 to the red teaming experiment.
> >
> > **Moreover, the first case study suggests to use alternate fairness metrics, showing that they are more successful. This feels a bit to me like re-inventing the wheel, as there are already other evals which are specifically tailored for measuring fairness, and my guess would be that such evaluation scores are already more correlated with actual fairness of outcomes in the setting that is tested. Another smaller qualm with the setup is that there were no error bars reported in Figure 2, and the number of resumes generated was quite small, when considering that they are AI generated, so the additional generation costs are minimal. This makes it hard to assess the significance of results.**
> >
> > As noted above, we do agree with the reviewer that it would be interesting to explore whether other technical notions of fairness might be correlated or not with these context-dependent outcomes, but we believe such studies are outside the scope of our paper.
> >
> > We recognize that the size of our dataset is relatively small; however, we note that all resumes were summarized 4 times (with different names) by each of 5 models, and then each of those were scored by a 70B parameter model, meaning that the overall compute costs for this experiment (or those across the whole paper) are not low (at least with our academic GPU resources).
> >
> > In response to the concern regarding error bars, we have added standard error bars to plots wherever possible (Figures 2 (left), 4, 5, 6, 8).
> >
> > **Despite the second case study being well executed, the takeaways are a little underwhelming – as mentioned above, I think they are already the standard practice for any entity that knows what they're doing.**
> >
> > We feel that it may not be fair to assume that any organization with sufficiently advanced ML researchers could anticipate all of these or other undemonstrated shortcomings of existing fairness techniques.  However, even if this were the case, we hope that the techniques themselves might in the future be built to meet the needs of deploying organizations without such personnel, of which there will likely be many.
> >
> > **I had trouble understanding the third case study, especially the experimental setup. How is the interaction history created exactly? Do you have an example of what an interaction looks like somewhere?**
> >
> > We have updated the main paper and appendix to clarify the details of the multi-turn experiment, including an example of an interaction history in the appendix (Table 9).  To clarify, an interaction history is created from a set of (input query, LLM response) pairs, where the input query comes from the GSM8K or MedQuad dataset and the LLM response is generated by Gemma-2-9B-instruct.

---

> > > ### Author Response · Authors · 2024-11-22
> > > **Author Follow-up**
> > >
> > > Hello, as the discussion period will be ending in a few days, we wanted to follow up and see if there are any remaining questions we can answer or any other changes we can make to address the reviewer’s concerns.  Otherwise, we hope that the reviewer may consider raising their score based on the clarifications we have provided with respect to novelty, and the requested modifications that we have made to the submission.  Thank you again for your feedback.

---

> > > > ### Comment · Reviewer_LRTp · 2024-11-24
> > > >
> > > > Thank you for your comments. I appreciate the new results, adding error bars, and adding experimental details.
> > > >
> > > > I'm broadly still unconvinced about the value of the first two case studies.
> > > >
> > > > One of the main takeaways of the first experiment IIUC is a call to action to build suites of fairness evaluations/benchmarks which are context specific and tied to the deployment scenarios in question. This is the main thing that would help "meet the needs of deploying organizations without such personnel", as long as they were aware of such efforts. That said, building more benchmarks and evaluations already seems to be a mainstream effort – indeed, in my opinion, this is mostly because researchers are already aware of the limitations of single benchmark scores which are not context-specific. Regardless of the reasons why, the fact that building more context specific evaluations and benchmarks is already a significant area of effort makes it unclear what exactly the contribution of this paper is.
> > > >
> > > > Similarly, even the second case study feels a bit like a "call to action" for an action which is already well-underway.
> > > >
> > > > Because both case studies are somewhat shallow (having more depth be "beyond the scope of the paper"), the call to actions are also less detailed, reducing their novelty relative to what people already are acting upon.
> > > >
> > > > Generally, I'm left with a sense that the paper could have been better executed as a position paper or literature review which focuses more on where the field is already moving towards (a multitude of context-specific benchmarks, which will invariably be somewhat different from actual deployment scenarios), and whether _that_ will be sufficient. The one thing that seems harder to fit into that narrative is the last case study, which seems more novel.
> > > >
> > > > ---
> > > >
> > > > In light of the above, I'd like to maintain my score for now.

---

> > > > > ### Author Response · Authors · 2024-11-26
> > > > > **Author response**
> > > > >
> > > > > We thank the reviewer for their further consideration of our work, and for responding to our rebuttal.
> > > > >
> > > > > First, we would like to respond to the point about context-specific evaluations.  We agree that researchers are already aware of the limitations of single benchmark scores which are not context-specific.  However, taking up the general goal of context-dependence does not guarantee that future benchmarks will be any better for serving regulatory purposes than existing ones.  In particular, the novelty in our case study is highlighting how existing fairness research is not useful because it is focused on models making allocative decisions, which is well-covered by existing law.  For example, [1] and [2] are both very recent papers (last ~6 months) on fairness in LLM-assisted hiring applications.  However, because the LLM under inspection acts as a decision maker, these works produce tools that are orthogonal to the LLM summarizer application in our case study.  Without an explicit call to change this direction, it seems likely that researchers will continue to double down on tools to support traditional discrimination law, while ignoring the needs of emerging regulation.  We will revise the paper to explicitly contrast our case study to these recent works and thus refine this point.
> > > > >
> > > > > Further, we maintain our claim to novelty with respect to our case study on the potential for popular red teaming approaches to produce variable results.  We have performed an extensive literature search of extremely recent work (cited extensively in Section 4.2), and found no paper exhibiting similar experimental results or conceptual arguments.  Though we agree that researchers are aware that red-teaming has shortcomings, we once again stress the importance of knowing the specific shortcomings that they should tackle in future research, if the goal is to support real anti-discrimination enforcement.
> > > > >
> > > > > Overall, we hope that the reviewer considers our work not as generally saying “evaluation is brittle”, as we agree that this is common knowledge.  Instead, our work acts as the first study to pick out the particular points of brittleness that act as bottlenecks for enforcing emerging GenAI discrimination policy.  We believe that each of our case studies serve to illustrate this purpose, and that they are all built upon perspectives that stem from our central thesis, and thus are not taken in other previous work.
> > > > >
> > > > > Thank you again for engaging, and if there are any other points of clarification or questions you  have, we are happy to answer.
> > > > >
> > > > > [1] Auditing the Use of Language Models to Guide Hiring Decisions https://arxiv.org/abs/2404.03086
> > > > > [2] Gender, Race, and Intersectional Bias in Resume Screening via Language Model Retrieval https://arxiv.org/abs/2407.20371

---

> > > > > > ### Comment · Reviewer_LRTp · 2024-12-01
> > > > > >
> > > > > > I'd like to thank the authors for their continued engagement.
> > > > > >
> > > > > > Unfortunately, after spending some more time on the submission, I still feel quite on the fence about it.
> > > > > >
> > > > > > On the one hand, I agree with Reviewer 79Lv that "Mapping legal concepts to technical concepts is the main novelty of this paper.". That said, I don't believe I have the expertise to evaluate the correctness of the connections to the legal literature. Moreover, that specific contribution may not fall as squarely in ICLR (AFAIK), and would maybe be more appreciated in a venue like FAccT.
> > > > > >
> > > > > > When evaluating the experiments and the takeaways from them, while I agree with you that your work goes beyond simply saying that "evaluation is brittle", I also agree with Reviewer LRTp that there is still some lack of depth in the experiments. Moreover, I do think the gap between the setting you consider and real-world scenarios make the takeaways from your case studies lose some force.
> > > > > >
> > > > > > As a side note, I would suggest summarizing the takeaways from each case study in a Conclusion or Discussion section – I think that can make more clear what your main claims are, how novel they are, and how well-supported they are by your experimental results.
> > > > > >
> > > > > > I'd like to maintain my score, but I'm open-minded to change it during the discussion with other reviewers.

---

> > > > > > > ### Author Response · Authors · 2024-12-02
> > > > > > >
> > > > > > > We thank the reviewer for their response, and continued consideration of our submission.
> > > > > > >
> > > > > > > The ICLR call for papers explicitly includes topics such as “societal considerations including fairness, safety, privacy.” Our work connects technical phenomena in generative AI with anti-discrimination regulation, which has historically been a cornerstone of societal efforts to address unfair practices like redlining (Fair Housing Act of 1968) and hiring discrimination (Title VII). By addressing generative AI discrimination risks proactively, our paper fills a critical gap at a pivotal moment. As the development and enforcement of emerging GenAI frameworks increasingly relies on technical expertise, aligning technical and regulatory considerations is essential for society to achieve effective outcomes.
> > > > > > >
> > > > > > > We recognize that vetting legal arguments in an ML conference review may pose challenges. However, we believe this presents a significant Catch-22 for the ML research community that must be overcome: papers addressing regulatory efforts may struggle to find acceptance because reviewers lack the tools to evaluate their significance or correctness. Yet, this very knowledge is essential for ML researchers to engage effectively with anti-discrimination enforcement and ensure the success of such efforts in critical AI applications. Our paper stands out because it grapples with concrete societal concerns that are often overlooked in ML fairness research. By doing so, we aim to help normalize the integration of these crucial considerations into the field, .
> > > > > > >
> > > > > > > Thank you again to the reviewer for the time and effort taken on our paper, it is very much appreciated!

---

### Official Review · Reviewer_8bPr · 2024-11-07

**Soundness:** 2
**Presentation:** 2
**Contribution:** 2
**Rating:** 3
**Confidence:** 4

**Summary:**

This paper proposes that traditional (1) fairness evaluations and (2) popular red teaming approaches are not effective in their current state for generative AI evaluations, arguing that these evaluations should be contextualized to a given domain rather than defined relatively a priori for (1) and focus on "producing standardized and robust attack frameworks" for (2). The arguments for (1) are themselves not novel (indeed, the authors cite commentary from the White House OMB to this end). Instead, the paper is intended to highlight a sample of cases where the outcome maps true to the community's general intuition. They consider four empirical evaluations on1) summarizing resumes through LLMs,  2) variability in red teaming for offensive screening, 3) how single turn and multi turn interactions should be evaluated differently, and 4) how user modification influences model outputs. Each case's methodology largely focuses on prompting existing models then evaluating the outputs.

**Strengths:**

This paper is grammatically well-written, and provide coverage of important recent regulatory discussion motivating the work. The authors touch on a number of important questions. The discussions, specifically around red-teaming and single vs multi turn interaction, were interesting, though I am less familiar with that body of literature on red-teaming and so rely on other reviewers to discuss its relevance and novelty. The paper is timely and thus of interest to ICLR's audience.

**Weaknesses:**

I have a number of concerns with this work that lead me to believe that it is not ready for publication. I will speak to the broadest concern first, then focus on specifics of experimental protocol second.

Paper structure. This paper is quite broad in its application. The case studies are relatively shallow, as a result, and connecting the themes together weakens the contribution. For example, I believe that the experiments and discussion regarding red teaming and the single vs multi-turn are contributions that should, when done well, be published in a venue like ICLR, ICML, or FACCT. But the experimental protocols, the appendices, and the discussion do not currently do these concepts justice. I would suggest the authors either revisit how they synthesize this work into one coherent story, or split into separate papers where they more deeply engage with the topics both experimentally and in takeaways for the community. This cannot be done in the time frame of a review cycle, but makes for much better papers.

Experiments. I have a number of questions, and concerns for the experimental study.

Experimental Setup for E1
This experiment relies on generated resumes, where different names based on ethnicity are included. The appendix is very sparse for these experiments, as are the main text explanations, which leads me to the following concerns:

(1) how were the resumes generated? In the same way that summarizing can embed bias so too can the generation be a result of samples that are not representative of our ideal. Specifically, how did the authors ensure that the resumes generated were close enough to or the same across each variable (e.g. ethnicity) s.t. the results were not confounded? Without thorough control for externalities, it’s unclear if the resulting summaries are actually indications of what the authors claim. Further, it would be useful if the authors repeated the results across several models. While the authors used separate models to produce and evaluate the resumes (which is a great step), this fact on its own is not enough to ensure reliable outcomes. Beyond this, it would be helpful if the authors include full details of the prompting, the model versions that they used to prompt (assuming they used APIs), along with the resulting resumes for reproducibility.

(2) I’m generally confused at how this experiment was structured, as it seems like there is a generated resume, a summary, and then a score, but that pipeline isn’t clear from the main paper or the appendix.

(3) I am also concerned that Llama is not a good proxy for this setting of resume evaluations per se. There’s a difference between summarizing the content of the resume and grading it. The description in the appendix includes the line, “In order to simulate interview decisions, we prompt Llama-3-70B to score each candidate 1-10 based on the summary of their resume, where a score of 9 or greater results in an interview.”  Summaries from language models make sense for LLM applications, but grading criteria is more likely to be applied on top of these summaries as either a classifier or some other learned (specialized) tool. A reasonable assessment is to look at how the summaries themselves are biased, resulting in the next step (scoring) being unfair. But it appears to instead be the case that the authors used generative tooling for each step. This introduces many degrees of freedom in the result that I would like to see minimized if possible. While the authors explicitly state that realism is not their focus, the descriptions of these experiments as they stand are not sufficient, and further detail is needed.

Experimental Setup for E2
This experiment involves using a RedLM (there are apparently 7) to generate question templates. The blanks in these templates are then filled (unclear how) and fed into the LLM of interest. In this way, the LLM of interest is then evaluated by its responses in aggregate, which are measured for toxicity etc. I think the experiment is being compared to a baseline measure of fairness/bias, but the baseline is not described. In other words, I do not understand how this experiment is intended to show that variability is a concern ( though I buy the premise). Further, there are two instances where the RedLM and the LLM of interest are the same... is there a reason why it's okay here to generate and then evaluate using said content? I don't know that in this specific case it is too much of an issue, but it is odd from an empirical perspective.

Experimental Setup E3
Again, this experimental idea is interesting. Changing conversation length is a nice and straightforward variable that I can imagine motivating a number of exciting new evaluations in the future. However, the actual experiment is not clear. For example, how are the attack "successes" measured? In general, I could not understand what was actually measured or what was changed from what was included in the writing.

Ultimately, the information shared in this paper is not enough to reproduce the experiments as-is. Each case study would benefit from both greater description, and deeper exploration. The authors are on a promising track and would benefit from an additional number of months to refine both studies and their communication of the issues at hand. I would like to see additional discussion of how to build on these concepts. I'm excited to see future iterations!

**Questions:**

In general, please see above for questions.

---

> ### Author Response · Authors · 2024-11-19
> **Author Response**
>
> We thank the reviewer for the time and care taken in reviewing our submission.  We are encouraged to see that they recognized the timeliness of our study and its interest to the ICLR audience, and felt that our case studies offered some interesting insights.  We would like to respond to particular feedback that the reviewer offered.
>
> **Paper structure. This paper is quite broad in its application. The case studies are relatively shallow, as a result, and connecting the themes together weakens the contribution. For example, I believe that the experiments and discussion regarding red teaming and the single vs multi-turn are contributions that should, when done well, be published in a venue like ICLR, ICML, or FACCT. But the experimental protocols, the appendices, and the discussion do not currently do these concepts justice.**
>
> The goal of our paper is to show how gaps between existing and emerging regulation and current fairness techniques can lead to deployment of reportedly fair, yet actually discriminatory GenAI systems. In doing so, we hope to highlight particular research directions, of the many available to GenAI researchers, that would actually support real-world efforts to enforce anti-discrimination in GenAI deployments. We appreciate the reviewer highlighting that individual case studies offer the potential to become interesting research studies in and of themselves; our hope is to spur such studies.  However, we believe that it is important to first give GenAI researchers the general direction necessary in order to support emerging regulatory efforts at achieving fairness in real applications in critical domains, and we do not know of any previous study offering such direction.  In response to this concern, we have clarified this point at the beginning of Section 4.
>
> **Experiments. I have a number of questions, and concerns for the experimental study.**
>
> We appreciate the reviewer’s concern regarding clarity of experimental methods.  In response, we have:
> - Updated the main paper and appendix to thoroughly explain all experimental details for each of the 4 case studies.  These details include all prompts, model versions, and hyperparameters necessary for reproduction.
> - Added a new Figure 7 illustrating the hiring experiment, and an updated written explanation of our data creation and overall experiment pipeline for experiment 1.
> - Included more thorough explanation and set of details for the red-teaming, multi-turn conversations, and image generation experiments.
> - Clarified our metrics, in particular for the red-teaming and multi-turn experiments.
>
> **Experimental Setup for E1 This experiment relies on generated resumes, where different names based on ethnicity are included. The appendix is very sparse for these experiments, as are the main text explanations, which leads me to the following concerns:
> (1) how were the resumes generated? In the same way that summarizing can embed bias so too can the generation be a result of samples that are not representative of our ideal. Specifically, how did the authors ensure that the resumes generated were close enough to or the same across each variable (e.g. ethnicity) s.t. the results were not confounded? Without thorough control for externalities, it’s unclear if the resulting summaries are actually indications of what the authors claim.**
>
> Regarding confounders in the resume generation process, we note that the categories of traits that we give to GPT4 do not include race, ethnicity, or highly related characteristics like religion or language.  In response to this concern, we have clarified this point in line 259.
>
> **Beyond this, it would be helpful if the authors include full details of the prompting, the model versions that they used to prompt (assuming they used APIs), along with the resulting resumes for reproducibility.**
>
> In response to the reviewer’s general concern about lack of details for this experiment, we have added all details necessary to reproduce this experiment to the appendix.
>
> **(2) I’m generally confused at how this experiment was structured, as it seems like there is a generated resume, a summary, and then a score, but that pipeline isn’t clear from the main paper or the appendix.**
>
> To clarify our experimental setup, we have added a new Figure 7 illustrating the data creation pipeline and overall experiment setup, and an updated written explanation.

---

> > ### Author Response · Authors · 2024-11-19
> > **Author Response (continued)**
> >
> > **(3) I am also concerned that Llama is not a good proxy for this setting of resume evaluations per se. There’s a difference between summarizing the content of the resume and grading it. The description in the appendix includes the line, “In order to simulate interview decisions, we prompt Llama-3-70B to score each candidate 1-10 based on the summary of their resume, where a score of 9 or greater results in an interview.” Summaries from language models make sense for LLM applications, but grading criteria is more likely to be applied on top of these summaries as either a classifier or some other learned (specialized) tool…While the authors explicitly state that realism is not their focus, the descriptions of these experiments as they stand are not sufficient, and further detail is needed.**
> >
> > In response to concerns about the LLM decision maker, we have performed new experiments in order to understand whether the LLM decision maker itself is the source of the bias in the selection rates shown in Figure 2.  To do so, resumes are summarized without an applicant’s name by Llama-2-7B, and then fed to the decision maker with stereotypical names from each of 4 groups. Selection rate results are shown in Figure 8, showing the decision maker to be significantly less biased when Llama-2-7B produces race-blind summaries, demonstrating that the main source of discrimination is likely the summarization model.
> >
> > Also, with respect to the appropriateness of simulating these decisions with Llama-3-70B, we emphasize that our experiment is not meant to be a high-fidelity simulation of a real world application, but instead to demonstrate a core tension between GenAI bias evaluation and discrimination regulation.  We also note that using generative models for evaluation is an increasingly common paradigm.  Overall, we believe the inclusion of the LLM decision-maker facilitates a more complete illustration of the issue at hand.
> >
> > **Experimental Setup for E2 This experiment involves using a RedLM (there are apparently 7) to generate question templates. The blanks in these templates are then filled (unclear how) and fed into the LLM of interest. In this way, the LLM of interest is then evaluated by its responses in aggregate, which are measured for toxicity etc. I think the experiment is being compared to a baseline measure of fairness/bias, but the baseline is not described. In other words, I do not understand how this experiment is intended to show that variability is a concern ( though I buy the premise). Further, there are two instances where the RedLM and the LLM of interest are the same... is there a reason why it's okay here to generate and then evaluate using said content? I don't know that in this specific case it is too much of an issue, but it is odd from an empirical perspective.**
> >
> > We would like to clarify specific points about the red teaming experiment for the reviewer.  In this case study, we are highlighting how red teaming, as prescribed by an increasing number of policy documents, might produce variable results, where the appearance of discrimination in red teaming is highly sensitive to the choice of red team (or underlying technique, model, hyperparameters, etc.).  The idea is that a deploying organization will have a set of models that they consider, and want to show regulators or auditors that among these models, they have chosen one that is relatively non-discriminatory.  We examine how degrees of freedom available to a red team using existing, highly-cited attack frameworks may enable them to produce a report with arbitrary fairness results, intentionally or not.  Though anecdotal, we have been told by members of a sophisticated industry research lab that they are encountering difficulty in reproducing and stabilizing the results of a recently-published, high-cited attack framework.  We thus feel that showcasing this in our work is a meaningful contribution to the bias testing literature.
> >
> > With respect to generating and evaluating from the same model, this was the case in the original Perez 2022 paper.  Also, we think it is reasonable that attacks might come from a different model, for example if a set of attacks is produced once and reused over time.  Thus we consider both scenarios.  In response to this concern, we have added clarification of this point in writing in Section 4.2.

---

> > > ### Author Response · Authors · 2024-11-19
> > > **Author Response (continued)**
> > >
> > > **Experimental Setup E3 Again, this experimental idea is interesting. Changing conversation length is a nice and straightforward variable that I can imagine motivating a number of exciting new evaluations in the future. However, the actual experiment is not clear. For example, how are the attack "successes" measured? In general, I could not understand what was actually measured or what was changed from what was included in the writing.**
> > >
> > > For the multi-turn experiment 3 metric, a successful attack is when the toxicity score of the response to a red teaming prompt is above the threshold.  We have updated the paper at line 462 to clarify this.
> > >
> > > **Ultimately, the information shared in this paper is not enough to reproduce the experiments as-is. Each case study would benefit from both greater description, and deeper exploration. The authors are on a promising track and would benefit from an additional number of months to refine both studies and their communication of the issues at hand. I would like to see additional discussion of how to build on these concepts. I'm excited to see future iterations!**
> > >
> > > We hope that we have sufficiently addressed the reviewer’s concerns with respect to the usefulness of our scope, and that we will be able to present a camera-ready version that contains all of the necessary information in order to appreciate our experimental results and overall contribution.  If there are any remaining questions we can answer for the reviewer, or any other changes we can make to strengthen our submission, we would be happy to do so.

---

> > > > ### Author Response · Authors · 2024-11-22
> > > > **Author follow-up**
> > > >
> > > > Hello, as the discussion period will be ending in a few days, we wanted to follow up and see if there are any remaining questions we can answer or any other changes we can make to address the reviewer’s concerns.  Thank you again for the time and consideration.

---

> ### Comment · Reviewer_8bPr · 2024-11-27
>
> The efforts taken by the authors to update the appendix (and to some extent the main paper) are great step towards responding to many of my concerns regarding replicability and clarity. The authors' review responses were helpful in explaining both methods and their intention. Further, the probing experiment on discriminatory summarizations is a nice addition (though missing confidence intervals). I will just briefly note that it would be helpful to not need to switch back and forth between main and appendix figures to piece together the comparisons described in the appendix.
>
> However, I was underwhelmed by the changes to the main body of the paper. My broader criticisms of the paper structure still stands, and this perspective seems to be reflected by other reviewers. As I mentioned in my original review, I believe this represents an interesting collection of ideas but requires iteration to refine the authors' narrative through additional scoping and discussion.
>
> On page 4, the authors took care to add a lengthy disclaimer regarding the mismatch between experiments to the real world applications. While this is an important consideration, it is ultimately not my concern--toy experiments are common, useful mechanisms for illustrating concepts; I remain unconvinced by the experimental depth across the case studies.
>
> Thus, while I will update the soundness score, my recommendation for reject still stands.

---

> > ### Author Response · Authors · 2024-12-03
> >
> > Thank you to the reviewer for considering our rebuttals.  We are glad that we were able to respond to many of your concerns, and that you found our responses helpful.  We appreciate the concern regarding the amount of contents in the appendix, and would make an effort to use the space in the main paper more effectively before a camera-ready.  Also, we politely note that Figure 8, with the probing experiment results, does have error bars.  We recognize that they may be thin and difficult to notice, and we will make them more visible in an updated version of the paper.
> >
> > We agree with the reviewer that the strength of the paper is its coverage of important recent regulatory discussion motivating the work, and that it touches on a number of important questions.  We also agree that these case studies could be extended into standalone research directions, and we are excited by that promise.  However, the goal of our work is to connect technical phenomena in generative AI with anti-discrimination regulation, a cornerstone of societal efforts to address unfair practices in important domains like medicine, housing, lending, and education.  By addressing generative AI discrimination risks proactively, our paper fills a critical gap at a pivotal moment in the history of this technology.
> >
> > We argue that as the development and enforcement of emerging GenAI frameworks increasingly relies on technical expertise, aligning technical and regulatory considerations is essential for society to achieve effective outcomes.  We believe that our work is an important first step in achieving this alignment, and that providing ML researchers with the knowledge necessary to support real anti-discrimination efforts in their work is a more important goal than individual research efforts along one or more of the axes we highlight.
> >
> > We thank the reviewer again for the time taken in reviewing our paper, and hope that they may consider our work in light of the potential larger societal impact, as opposed to its potential to produce incremental technical progress.

---

### Author Response · Authors · 2024-11-19
**Author Response**

We thank the reviewers for their time and consideration in offering valuable feedback on our submission.

First, we are encouraged to see that all reviewers appreciate the novelty and potential significance of our work in connecting the legal and technical literature around GenAI bias evaluation and demonstrating areas of misalignment.  Through four case studies, we are able to show how the gap between fairness testing techniques and regulatory goals can result in discriminatory outcomes in real-world deployments, a significant concern given the proliferation of these models in risk-sensitive domains like hiring and medicine.  We appreciate the recognition of the importance of the questions we are considering, the originality of our case studies, and that this subject is timely and of interest to the conference audience.

We also understand that there are concerns regarding the robustness of particular experimental findings within our case studies.  We would like to emphasize that our experiments are not meant to argue for particular fairness methodology or evaluation techniques.  Rather, they are meant to demonstrate how gaps between regulation and methodology can lead to situations where an actually discriminatory GenAI system is deemed sufficiently unbiased for deployment.  By doing so, we aim to highlight research directions, of the many available to GenAI researchers, that would actually support real-world efforts to enforce anti-discrimination in GenAI deployments.  To address these concerns, we have added such a statement explicitly in the first paragraph of section 4.

While their primary purpose is to illustrate the gaps between the legal and technical landscapes, we do feel that there is some important novelty in our experiments and results.  For example, while previous work has looked at LLM-based hiring pipelines, they have focused on cases where the LLM is the decision maker, which can be easily mapped to traditional discrimination law.  The originality in this experiment lies in considering an LLM-assisted hiring application where emerging AI regulation is needed in order to prevent discrimination, supporting the central theme of our paper.  In addition, though the results in Sections 4.2-4.4 may not “surprise” advanced GenAI researchers, we do not know of any previous work specifically exploring variability of red-teaming results, the effects of multi-turn interactions on fairness and toxicity, or the effects of hyperparameter choices on text to image bias.

We recognize the important concerns of multiple reviewers about the clarity and thoroughness of our experimental presentation.  In response, we have made significant changes to the submission.
- We have updated the main paper with more details where space permits, and clarified writing.
- We have significantly expanded and refined the experiments section in our appendix.  The goal of our new appendix is to ensure reproducibility from the paper itself, beyond the code that we have shared with reviewers and will release online.  We have included all relevant prompts, hyperparameters, and other details, and clarified the writing throughout so that experimental design can be more easily understood.
- We have added a new Figure 7 illustrating the hiring experiment
- **We have performed a new experiment probing the fairness of our LLM decision-maker** in order to understand how much discrimination may be coming from the summaries.
- We have added standard error bars to plots wherever possible (Figures 2 (left), 4, 5, 6, 8).

Please find that the PDF of our submission has been updated with the above noted changes, and others noted below in responses to each reviewer.  Thank you again.

---

### Meta-Review · Area_Chair_LThh · 2024-12-20

**Metareview:**

This paper looks at Generative AI regulation, specifically, bias evaluation (fairness testing) and the gap between techniques and regulatory goals. This is a particularly relevant topic currently (and it would be nice to see more papers on such topics at conferences like ICLR), which I think is a key strength / important aspect when reviewing this paper. It is also a very difficult topic to broach, as it is two different fields with different languages, and there are not many interdisciplinary conversations at places like ML conferences like ICLR.

Reviewers generally engaged deeply with this paper, and there was also a good discussion between reviewers and authors. Overall reviews vary. Strengths brought up by reviewers include the relevance and importance of the topic, and that the paper is well-written / easy to read. I also think the paper is stronger after including additional experimental details are requested by Reviewer 8bPr.

However, I find some key weaknesses in this paper, as brought up by reviewers too. Specifically, the case studies are (i) not connected thematically, (ii) not specific enough, and (iii) do not have clear takeaways. In my experience, specificity and thematic links are crucial when linking legal frameworks to technical research. As a major contribution of this paper is to link methods to policy, this becomes an important weakness. I encourage the authors to perhaps work on the story and how the case studies fit into it (the intro, for example, is well-written, but seems to overpromise given the case studies). This may require restructuring the paper or dropping case studies for a future version.

**Additional Comments On Reviewer Discussion:**

Reviewer 8bPr asked for clarifications on experiments, which authors broadly addressed. The reviewer is also concerned about the case studies being not connected thematically, and not linking sufficiently to the paper goals. I agree with this assessment. Reviewer LRTp has similar concerns about the case studies: I think a lot of their concerns are coming from the lack of coherent story across case studies. Note that I disagree with Reviewer LRTp about ICLR maybe not being a good venue for such work. Reviewer 79Lv is positive about this work, recommending accept, and the discussion raised interesting points, such as about the EU AI Act. However, I found a lot of this discussion to be very high-level. This would be fine for a primarily ML paper, but as this paper is linking policy to ML (as its major contribution), I find it lacking. I also think that the recommendations made by the paper need to be more specific / actionable, and agree with other reviewers that the case studies are not sufficiently thought out / part of a coherent paper story.

---

### Decision · Program_Chairs · 2025-01-22

Reject